# Species-specific circular RNA circDS-1 enhances adaptive evolution in *Talaromyces marneffei* through regulation of dimorphic transition

**Xueyan Hu**[1], **Minghao Du**[2], **Changyu Tao**[3], **Juan Wang**[2], **Yun Zhang**[1], **Yueqi Jin**[1], **Ence Yang**[1,2]*

**1** Department of Medical Bioinformatics, School of Basic Medical Sciences, Peking University Health Science Center, Beijing, China, **2** Department of Microbiology & Infectious Disease Center, School of Basic Medical Sciences, Peking University Health Science Center, Beijing, China, **3** Department of Human Anatomy, Histology and Embryology, School of Basic Medical Sciences, Peking University Health Science Center, Beijing, China

* yangence@pku.edu.cn

## Abstract

Thermal adaptability is a crucial characteristic for mammalian pathogenic fungi that originally inhabit natural ecosystems. Thermally dimorphic fungi have evolved a unique ability to respond to host body temperature by shifting from mycelia to yeast. The high similarity of protein-coding genes between these fungi and their relatives suggests the indispensable but often overlooked roles of non-coding elements in fungal thermal adaptation. Here, we systematically delineated the landscape of full-length circRNAs in both mycelial and yeast conditions of *Talaromyces marneffei*, a typical thermally dimorphic fungus causing fatal Talaromycosis, by optimizing an integrative pipeline for circRNA detection utilizing next- and third-generation sequencing. We found *T. marneffei* circRNA demonstrated features such as shorter length, lower abundance, and circularization-biased splicing. We then identified and validated that circDS-1, independent of its parental gene, promotes the hyphae-to-yeast transition, maintains yeast morphology, and is involved in virulence regulation. Further analysis and experiments among *Talaromyces* confirmed that the generation of circDS-1 is driven by a *T. marneffei*-specific region in the flanking intron of circDS-1. Together, our findings not only provide fresh insights into the role of circRNA in fungal thermal adaptation but also reveal a novel molecular mechanism for the adaptive evolution of functional circRNAs derived from intronic mutations.

## Author summary

Thermally dimorphic fungi, which switch between mold-like and yeast-like forms depending on temperature, are major human pathogens. Their ability to thrive at host body temperature is critical for causing disease, yet how they evolved this trait remains poorly understood. While proteins are known players, non-coding elements like circular RNAs (circRNAs) have been largely overlooked. Here, we study *Talaromyces*

**Data availability statement:** All data used in this study were submitted to the NCBI Sequence Read Archive database (GEO; https://www.ncbi.nlm.nih.gov/geo) under accession GSE231909.

**Funding:** This work was funded by National Natural Science Foundation of China (https://www.nsfc.gov.cn/; 32170091 and 31970008 to Ence Yang). The funders had no role in study design, data collection and analysis, decision to publish, or preparation of the manuscript.

**Competing interests:** The authors have declared that no competing interests exist.

*marneffei*, a deadly fungus causing systemic infections, and uncover a novel circRNA, circDS-1, that drives its temperature-dependent morphological changes and even virulence in macrophage. By combining advanced sequencing technologies, we mapped the circRNA landscape of *T. marneffei* and identified circDS-1 as essential for promoting the transition to the pathogenic yeast form. Strikingly, circDS-1 production relies on a unique genomic region exclusive to *T. marneffei*, absent in closely related species. This region likely emerged during evolution to enable circRNA formation, potentially through interactions with RNA-binding proteins. Our findings reveal how non-coding RNAs contribute to fungal adaptation and pathogenicity, offering new insights into a hidden layer of genetic regulation. This work not only advances our understanding of fungal evolution but also opens avenues for targeting circRNAs to combat infections.

## Introduction

Although occupying only a minuscule fraction (<0.5%) of the vast spectrum of fungal diversity, human pathogenic fungi cause over 300 million severe infections with more than 1.5 million deaths annually [1–4]. A common feature of human pathogenic fungi is having successfully established heat tolerance, strategically overriding mammalian body temperature defense [5–8], when evolving from environmental fungi. A few fungal species, known as thermally dimorphic fungi, even evolve mechanisms to utilize elevated temperature as the trigger for morphogenesis and pathogengesis [9,10]. Thus, understanding how environmental fungi adapt to host body temperature is a critical puzzle piece in deciphering the adaptive evolution of fungal pathogenicity.

Thermally dimorphic fungi grow as saprophytic mycelia in environment, and shift to pathogenic yeasts when entering host body [5]. The dimorphic transitions between mycelia and yeasts are essential for their pathogenicity and transmission [6]. Moreover, the intimate connection between temperature change and morphological alterations renders thermally dimorphic fungi a pioneer model for unravelling molecular mechanisms of thermal adaptation in human pathogenic fungi [8]. Various protein-coding genes of thermally dimorphic fungi have been identified in orchestrating temperature response and morphological transition [6,11]. However, the roles and contributions of non-coding RNAs in adaptive evolution of thermally dimorphic fungi remain predominantly unexplored, despite their involvement in various biological processes in fungi [12,13].

Circular RNAs (circRNAs) are a prominent subtype of non-coding elements characterized by covalently closed continuous loops [14]. The special structure of circRNAs are formed through a process called back-splicing, where a downstream splice donor site is joined to an upstream splice acceptor site, largely facilitating by complementary sequences or RNA binding protein motif in flanking region and specific secondary structures [15–17]. This results in the formation of a circular structure instead of the typical linear mRNA, with the sequences of the circular and linear forms nearly identical except for the back-splicing junctions (BSJs). Accumulating evidence shows that circRNAs play widespread roles in shaping phenotypes and responses to environmental stimuli in plants, animals, and even bacteria [18–21], by acting as microRNA sponges, interacting with RNA-binding proteins and regulating transcription [14]. As circRNAs serve as important regulators of gene expression and cellular pathways across eukaryotes, we speculate that thermally dimorphic fungi may evolve circRNAs involved in thermal adaptation, although none of fungal functional circRNAs have been identified yet.

*Talaromyces marneffei* is a classical thermal dimorphic fungus that is closely related to the non-dimorphic fungus *T. stipitatus* [22], thus serving as an ideal subject for understanding circRNA roles in fungal temperature adaptation. To test our hypothesis, we firstly developed NreT-seq, a more powerful strategy for circRNA detection by combining the strengths of short-read and long-read sequencing technologies. Utilizing *T. marneffei* as a proof-of-concept, our investigation leverages NreT-seq to unveil condition-specific circRNA profiling under mycelial and yeast conditions, and found that circRNAs in *T. marneffei* have similar back-splicing and expression characteristics to higher eukaryotes. Then, we successfully identified and validated the first functional fungal circRNA which was named circDS-1, implicated in the manipulation of dimorphic transition and virulence in *T. marneffei*. Finally, we proved that the generation of circDS-1 was determined by *T. marneffei*-specific intronic region in its flanking intron, which might have been generated during the temperature adaptation of the *T. marneffei*. Over all, our study demonstrates a novel molecular mechanism of adaptive evolution, which is characterized by intronic variation driving generation of functional circRNA, in the temperature associated morphogenesis and pathogenicity of *T. marneffei*.

## Results

### CircRNA identification using NreT-seq in *T. marneffei*

Obtaining the complete sequence of circRNA is a prerequisite for fully understanding its potential functions and conservation [14]. Despite the exceptional accuracy of next-generation sequencing (NGS), it falls short in providing full-length circRNA sequences due to their overlap with cognate linear counterparts [23]. On the other hand, third-generation sequencing (TGS) enables the experimental determination of full-length circRNA sequences, albeit with relatively higher sequencing errors [24–26]. To systemic identify full-length circRNAs while decrease false positive, we established a re-correction strategy called NreT-seq (<u>N</u>ext-generation sequencing <u>re</u>-corrected <u>T</u>hird-generation <u>s</u>equencing) by integrating NGS and TGS (see Methods). We compared four methods, namely NGS-only, TGS-only, NGS&TGS (the intersection of NGS and TGS), and our NreT-seq (Fig 1A), utilizing a mouse dataset [26] with the circAtlas database as benchmark [27] as lacking corresponding fungal datasets. Although the NGS&TGS exhibited the best precision among full-length circRNAs, its lowest recall suggested possible systematic bias in circRNA identification. In contrast, NreT-seq demonstrated a balanced sensitive and precision when compared to the other full-length methods, highlighting its efficiency in integrating sequencing data for optimal information utilization (Fig 1B).

By employing NreT-seq in both mycelial and yeast conditions of *T. marneffei* (S1 Table), we identified a total of 3,891 high-quality circRNAs, approximately 60% of which were supported by both TGS and NGS (Fig 1C). Among the ten randomly selected circRNAs, back-splicing site and full-length sequence of nine circRNAs were validated by divergent primers, in which three top highly expressed were selected for Sanger sequencing (Figs 1D **and** S1), except for the one that failed in primer design, suggesting high accuracy of the identified circRNAs. Additionally, only 2 out of 9 validated circRNAs were identified in the results of NGS&TGS, indicating a potential identification bias associated with the NGS&TGS strategy in systematic screening, and highlighting the necessity of introducing NreT-seq.

### Sequences conservation of circRNAs in *T. marneffei*

Based on full-length sequences or back-splicing sites (BSJs) of the identified high-quality circRNAs, around 20% of protein-coding genes generating circRNAs with up to 27 circRNA isoforms, which is fewer than in human (up to thousands per gene) [24]. A significant portion (53.5%) of circRNA-deriving genes (CDGs) exhibited remarkable conservation within the

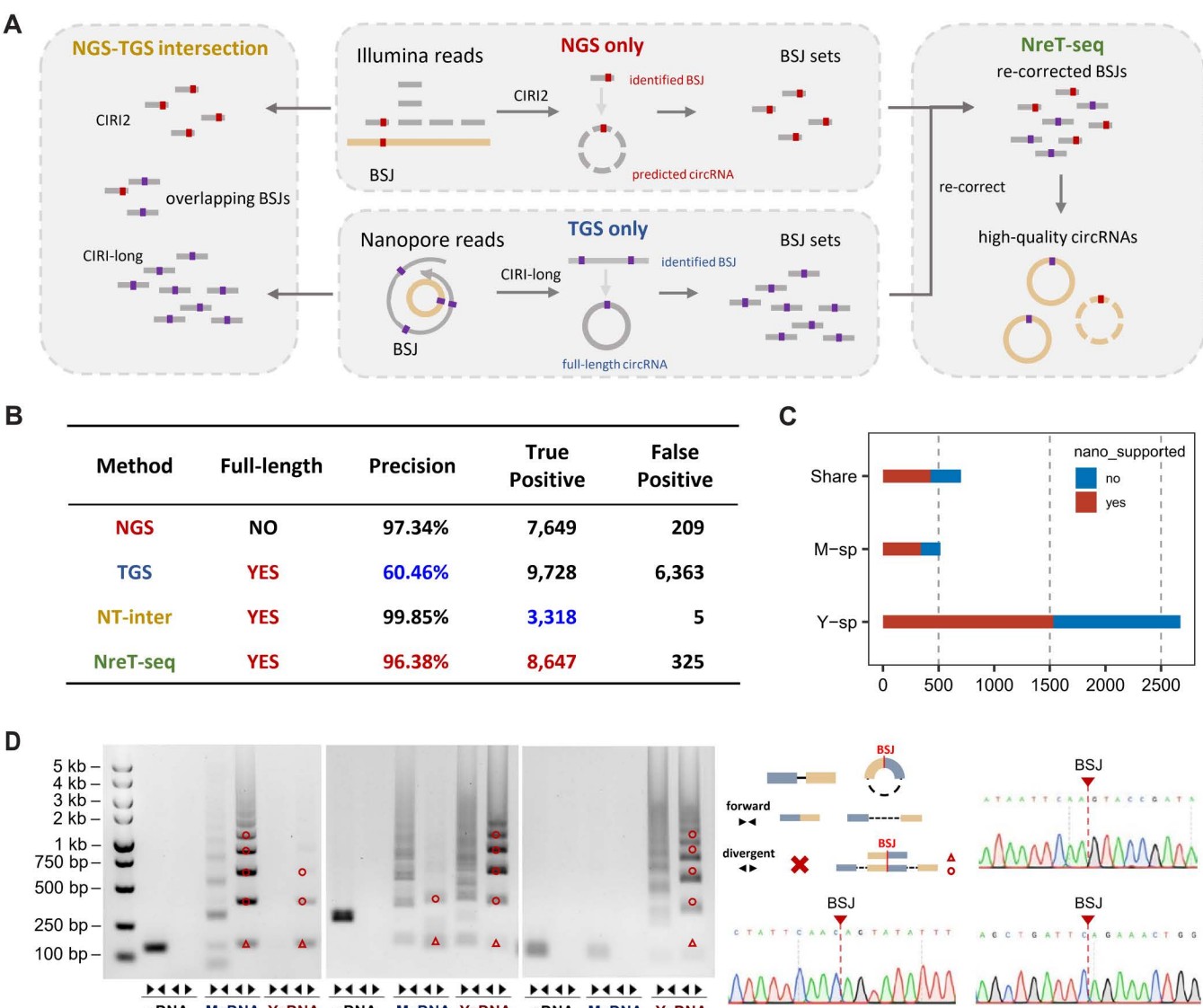

**Fig 1. The landscape of *T. marneffei* circRNAs. A**, Schematic diagram of four circRNA identification strategies. NGS-only circRNAs were identified by CIRI2 using Illumina sequencing reads. TGS-only circRNAs were identified by CIRI-long using Nanopore sequencing reads. NGS&TGS circRNAs were overlapping results of CIRI2 and CIRI-long. NreT-seq circRNAs were identified by re-correcting BSJs supported by Nanopore full-length reads using Illumina short reads. **B**, Comparison of the four identification strategies outlined in (A). **C**, Counts of *T. marneffei* circRNA identified by NreT-seq. M-sp presents mycelia-unique circRNAs; Y-sp presents yeast-unique circRNAs; share presents circRNAs identified in both mycelia and yeast. **D**, Agarose gel electrophoresis and Sanger sequencing of randomly selected circRNAs. Divergent primers could amplify BSJ of circRNAs, thus products could be detected in mycelia and yeast cDNA (McDNA and YcDNA) but not genomic DNA (gDNA). Forward primers were designed as controls to exclude trans rearrangement events at the genomic level that may mimic BSJ sequences. BSJ is represented by a red triangle, and the red circle represents the amplification product of "BSJ+full length*n", which is different from linear RNA.

*Talaromyces* genus, surpassing the conservation observed in non-circRNA-deriving genes (NCDGs) (Fisher's Exact Test, $P = 3.76 \times 10^{-56}$), which was consistent with human that some circRNAs originated from conserved genes [24].

Among the high-quality circRNAs, the full-length sequences of 2,291 *T. marneffei* cir-cRNAs were identified by TGS. These full-length circRNAs (fl-circRNAs) exhibited a length range of 44 nt to 1,904 nt with an average length 313 nt, in which 34.6% circRNAs were shorter than NGS predicted length (Fig 2A). We then annotated the exon/intron composition

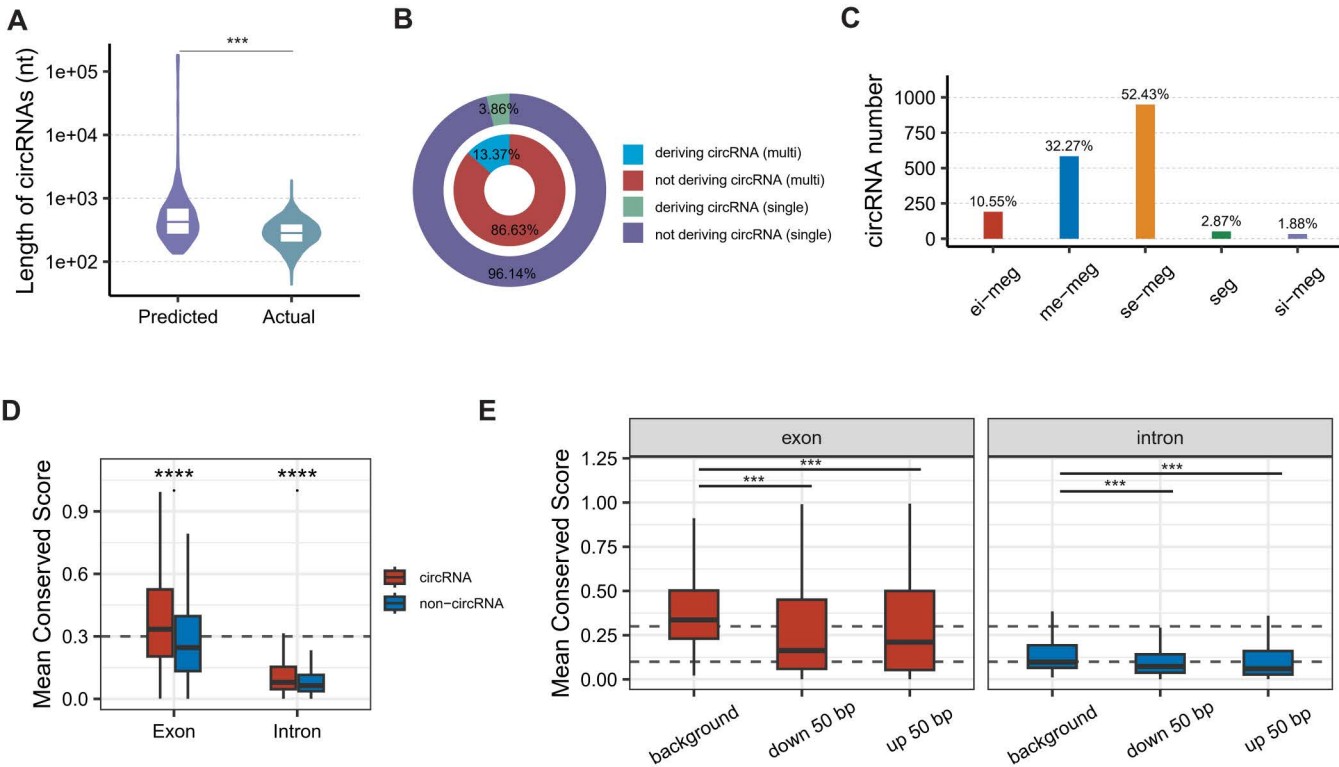

**Fig 2. The exon/intron usage and conservation of *T. marneffei* circRNAs. A**, Predicted and actual length distribution of all circRNAs. **B**, Circularization percentage of single-exon genes and multi-exon genes. **C**, Counts of five subtypes of circRNAs. Specifically, "seg-circRNAs" represent circRNAs derived from single-exon genes; "se-meg-circRNA" denotes circRNAs comprised of a single exon derived from multi-exon genes; "si-meg-circRNA" signifies circRNAs comprised of a single intron derived from multi-exon genes; "me-meg-circRNA" indicates circRNAs comprised of multiple exons derived from multi-exon genes; and "ei-meg-circRNA" denotes circRNAs comprised of exons and introns derived from multi-exon genes. **D**, Conservation comparison of circRNA exons and circRNA introns against corresponding background that consisted of exons or introns not used by circRNAs. Mean conservation scores were calculated using the PhastCons module of PHAST. **E**, Conservation comparison of circRNA flanking regions flanked by exons and introns is compared with background exons and introns that do not surround the circRNA, respectively. Mean conserved scores were calculated by PhastCons module of PHAST. The statistical significance of the comparisons was assessed using the Wilcoxon Rank Sum and Signed Rank Tests. Asterisks denote the level of significance: ***, $P$ value < 0.001; ****, $P$ value < 0.0001.

of fl-circRNAs according to their parent genes to analyze the exon and intron usage of circRNA. Similar to higher eukaryotic organisms, circRNAs preferred to be generated from multi-exon CDGs than single-exon CDGs (Fig 2B, Pearson's Chi-squared test with Yates' continuity correction, $P$ value = $3.13 \times 10^{-21}$), suggesting introns may be one of the factors promoting circularization. Unlike human [24], single-exon circRNAs from multi-exon CDGs were prevalent (52.4%), while multi-exon circRNAs that are dominant in human were less common (32.3%) in *T. marneffei* (Fig 2C). Similarly, the percentage of exon-intron circRNAs in *T. marneffei* was higher than in human (10.6% *vs.* < 4%) [24,28], which originated from significantly longer introns (Wilcoxon rank sum test with continuity correction, $P$ value = $4.11 \times 10^{-9}$), indicating a regulated rather than random retention mechanism in *T. marneffei*. This observation contrasts with the tendency for shorter introns to be retained in humans [28], suggesting potential differences in the splicing mechanism of circRNAs in *T. marneffei*, possibly due to the "intron definition" splicing model, as opposed to the "exon definition" splicing model in human with longer introns [29].

Upon exon and intron usage analysis, circRNA exons (Wilcoxon rank sum test, $P < 1 \times 10^{-314}$) and introns (Wilcoxon rank sum test, $P = 3.13 \times 10^{-164}$) exhibited significantly higher

conservation scores compared to their cognate linear counterparts within *Talaromyces* (Fig 2D), which is consistent with observations in mammals [27]. In contrast, a distinctive conservation pattern emerged in the flanking sequences surrounding circRNAs, exhibiting markedly lower conservation scores for both upstream and downstream flanking regions compared to their corresponding regions within *Talaromyces* (Fig 2E). Moreover, 5% of circRNAs were recognized as originating from non-conserved *T. marneffei*-specific flanking sequences. These *T. marneffei*-specific flanking sequences exhibited enrichment in functional categories associated with ATP synthesis and ribosomal processes (S2 Fig) that were essential in growth and development [30,31], suggesting a potential role of these circRNAs in distinctive biological processes or adaptation of *T. marneffei*.

## Screening of dimorphism associated circRNAs in *T. marneffei*

To identify whether circRNAs are involved in dimorphic transition regulation of *T. marneffei*, we firstly initiated differential expression analysis. A total of 373 circRNAs were identified as differentially expressed (DE) between mycelial and yeast conditions, comprising three mycelium-specific circRNAs and 357 yeast-specific circRNAs (Fig 3A). Different from the DE genes, which were predominantly associated with biogenesis and metabolic processes as previously reported [32], DE circRNAs displayed significant enrichment in translation, glycolytic processes, and nucleocytoplasmic transport processes (Fig 3B). Furthermore, circRNAs with *T. marneffei*-specific flanking sequences demonstrated a notable enrichment in DE circRNAs (Fisher's Exact Test for Count Data, $P = 0.006$), suggesting a potential link between species-specific flanking sequences and the regulatory functions of circRNAs during dimorphic transition of *T. marneffei*.

Since some circRNAs are co-expressed with parental genes, this poses a challenge in distinguishing functionally active DE circRNAs or merely bypass products. We further preformed differential transcription (DT) analysis to identify circRNAs that with differential expression pattern with their parental genes (see Methods). For example, the differential expression degree of circRNA is higher than that of its parental genes, lower than that of its parental genes, or is opposite to the differential expression trend of genes. Approximately 40% of the DE circRNAs were found to be significantly differentially transcribed (Fig 3C), suggesting potential post-transcriptional regulatory mechanisms during *T. marneffei* circRNA turnover. We then established the significance score of DT circRNAs and verified the back-splicing and differential expression via divergent primers of top 5 circRNAs (Figs 3D **and** S3). Among these validated circRNAs, circTM020485 generated by the exon 6 of gene *TM020485*, renamed by circDS-1, was selected for further exploration for the highest fold change of DT as well as flanked by non-conserved *T. marneffei*-specific sequences.

## circDS-1 involves in morphogenesis of mycelia and yeast

To validate the role of circDS-1 in dimorphic transitions, we firstly constructed mutants ectopically overexpressing circDS-1 that referred to as circDS-1^OE (see Methods). RT-qPCR confirmed that only circDS-1, but not its cognate linear transcripts, was significantly overexpressed (OE) in three individual circDS-1^OE biological replicates (#rep1-3) under both mycelial and yeast conditions (Fig 4A). Nevertheless, the expression of linear transcripts is slightly disturbed, possibly due to partial sequence similarity with circDS-1. Under 25°C, in contrast to the wild type (WT) strain that grew as mycelia and produced spores with the low circDS-1 expression level, the circDS-1^OE mutants exhibited obvious mycelial dysplasia and reduced sporulation under abnormal high circDS-1 expression levels (Fig 4A). Interestingly, under 37°C conditions, a temperature at which circDS-1 was highly expressed in the WT strains, the colony edge of the circDS-1^OE mutants with higher circDS-1 exhibited fine burr

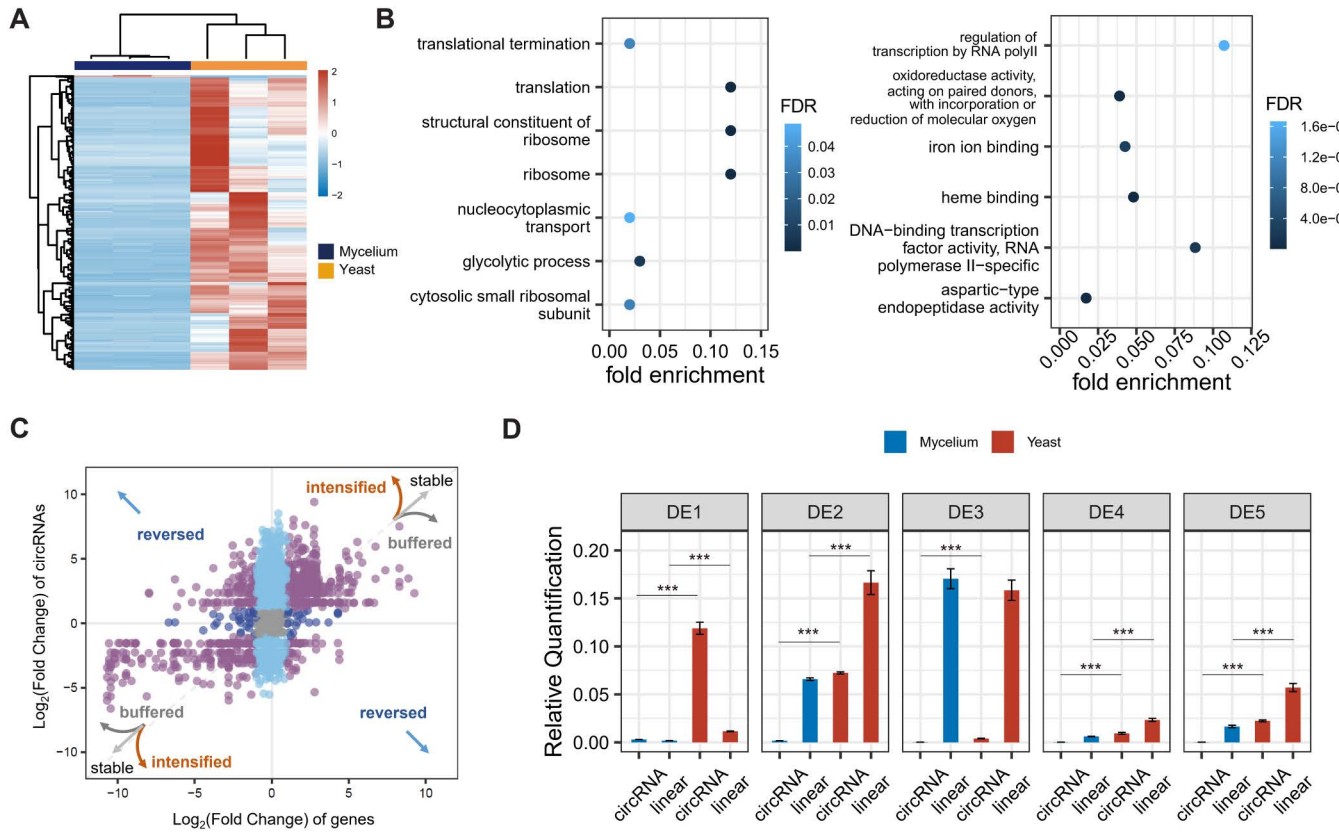

**Fig 3. Differential expression of circRNAs between mycelia and yeast. A**, Heatmap of up-regulated and down-regulated circRNAs, with the *z*-score normalized expression levels indicated by the color bar (red indicating a relative higher expression level and blue indicating a lower expression level). **B**, Gene Ontology enrichment of DE circRNAs (left) and DE genes (right). The shade of blue represents the FDR, and the Y-axis is the enriched GO term. **C**, Fold changes of differential expression (Yeast v.s. Mycelia) of circRNAs and their cognate linear transcripts. The "stable" shows that the differential expressed fold changes of circRNAs are same with their parent genes. The "intensified" shows that the differential expressed fold changes of circRNAs are larger than their parent genes. The "buffered" shows that the differential expressed fold changes of circRNAs are less than their parent genes. The "reverse" shows that the differential expressed fold changes of circRNAs are opposite with their parent genes. **D**, Differential expression validation of the top 5 circRNAs and their cognate linear transcripts under mycelium and yeast conditions with *actin* as the reference gene. Each qPCR experiment was performed with three technological replicates and RQ (Relative quantification) was calculated by $2^{-\Delta\Delta Ct}$ methods. The statistical significance of the comparisons was assessed using the Wilcoxon Rank Sum and Signed Rank Tests. Asterisks denote the level of significance: ***, *P* value < 0.001.

and dumped yeast cells compared with the smooth colony morphology of the WT strains (Fig 4B), indicating an excessive accumulation of yeast cells. These results demonstrated that circDS-1 could manipulate both mycelia and yeast morphogenesis under abnormal high expression.

To confirm whether circDS-1 participated in morphogenesis under physiological expression levels, we then generated circDS-1[RNAi] mutant strains using shRNA#1-3 spanning the BSJ sequences (see Methods). CircDS-1 was significantly knocked down (KD) with different efficiencies in yeast at 37°C, in which shRNA#1 (circDS-1[RNAi]#1) with BSJ deviation coordinates −11/+10 had the highest circDS-1 knockdown efficiency (Fig 4C). Due to the potential off-target properties of shRNA, cognate linear transcript of circDS-1 with almost similar sequences was inevitably affected slightly [33]. The surfaces of three individual circDS-1[RNAi] mutants exhibited dense pellets which was fill with multinucleate hyphae (>80%) while retaining a few normal yeast cells (Fig 4C), suggesting a potential occurrence of cell division inhibition. At 25°C where circDS-1 was extremely low expressed, slight morphological changes was observed in the three individual circDS-1[RNAi] mutants.

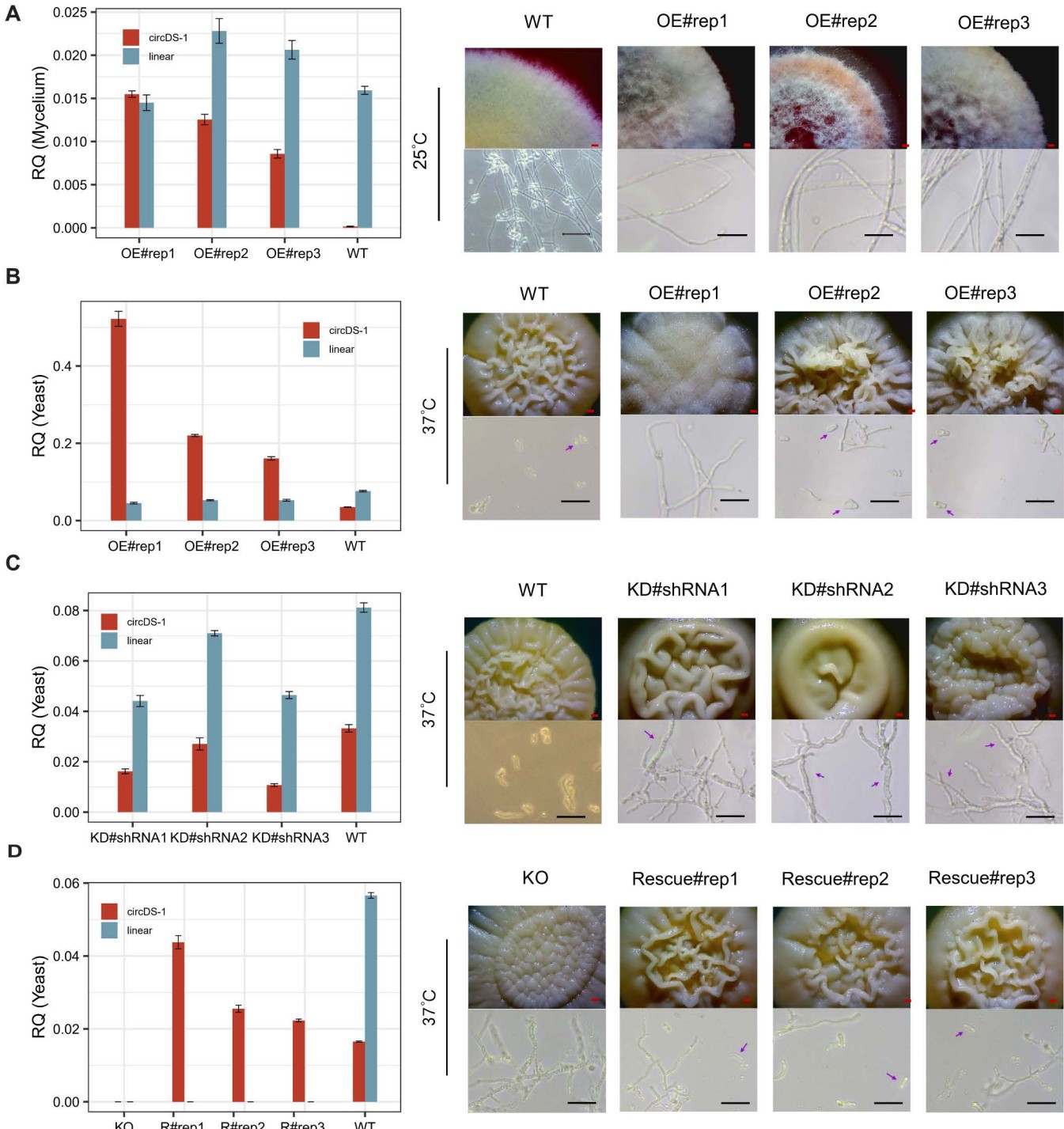

**Fig 4. circDS-1 modulates morphogenesis of mycelia and yeast. A,** Relative quantification (RQ) of circDS-1 and its corresponding linear transcript in cirscDS-1$^{OE}$ and wild type (WT) with *actin* as the reference gene under mycelia condition (left) and phenotypes of biological replicates of cirscDS-1$^{OE}$(OE#rep1-3) and WT (right). Each qPCR experiment was performed with three technological replicates and RQ was calculated by 2$^{-\Delta\Delta Ct}$ methods. **B,** Relative quantification (RQ) of circDS-1 and its corresponding linear transcript in cirscDS-1$^{OE}$ and wild type (WT) with *actin* as the reference gene under yeast condition (left) and phenotypes of biological replicates of cirscDS-1$^{OE}$(OE#rep1-3) and WT (right). Each qPCR experiment was performed with three technological replicates and RQ was calculated by 2$^{-\Delta\Delta Ct}$ methods. **C,** Relative quantification (RQ) of circDS-1 and its corresponding linear transcript in circDS-1$^{RNAi}$ and wild type (WT) with *actin* as the reference gene under yeast condition (left) and phenotypes of circDS-1$^{RNAi}$ generated by three different shRNA (#1-#3) and WT (right). Each qPCR experiment was performed with three technological replicates and RQ was calculated by 2$^{-\Delta\Delta Ct}$ methods. **D,** Relative quantification (RQ) of circDS-1 and its corresponding linear transcript in $\Delta TM020485$ and circDS-1 rescued $\Delta TM020485$ (Rescue#rep1-3) with *actin* as the reference

gene under yeast condition (left) and phenotypes of three biological replicates of circDS-1 rescued Δ*TM020485* (Rescue#rep1-3) and Δ*TM020485* (right). Each qPCR experiment was performed with three technological replicates and RQ was calculated by $2^{-\Delta\Delta Ct}$ methods. The purple arrows in the figures mark the typical morphological changes of strains. Scale bar of colony (red), 1 mm; Scale bar of cell (black), 20 μm.

We then generated the Δ*TM020485* mutant and circDS-1 rescued mutants to eliminate the influence of off-target effects in circDS-1 knockdown and confirm that the regulatory effect was specifically attributable to circDS-1 rather than its linear counterpart (see Methods). The Δ*TM020485* mutant was validated as lacking both circDS-1 and cognate linear transcript, exhibiting the similar phenotype as the above circDS-1 knockdown strains at 37°C (Fig 4D). On the contrary, the three individual circDS-1 rescued Δ*TM020485* mutants (Rescue#rep1-3) showed mostly normal surfaces and rescued normal yeast cells (>80%) (Fig 4D), indicating the down-regulated of circDS-1 could regulate morphogenesis of yeast cells independent to its cognate linear compartment.

## circDS-1 involved in dimorphic transition and virulence

To further figure out whether circDS-1 involving in dimorphic transition processes, we first quantified its expression over time and observed significant dynamic expression patterns during the wild-type dimorphic transition (S4 Fig). Specifically, circDS-1 exhibited a steady increase during the mycelia-to-yeast transition, while its expression initially increased before rapidly decreasing during the yeast-to-mycelia transition, suggesting that circDS-1 may have an early response function in morphological transition. Then we conducted morphological observations at different time points (from 0 h to 168 h) during both mycelia-to-yeast and yeast-to-mycelia transition. During the conversion of mycelia to yeast, the down-regulation of circDS-1 induced about 20% abnormal spore germination in both circDS-1 knock down and Δ*TM020485* strains but not in circDS-1 rescued Δ*TM020485* strain. As transition progresses, obviously inhibited the formation of arthrospores and reduced yeast cells were observed in circDS-1 knock down and Δ*TM020485* strains but not in circDS-1 rescued Δ*TM020485* strain, while the circDS-1 overexpressing strain had a certain yeast conversion-promoting effect (Fig 5A), demonstrating the down-regulation of circDS-1 inhibited mycelia-to-yeast transition. In the conversion of yeast to mycelia, the down-regulation of circDS-1 proceeded the hyphal growth slightly without influencing the final mycelia phenotype (Fig 5B), consistent with the observation that down-regulation of circDS-1 expression in mycelia did not affect colony growth. However, circDS-1 overexpression slightly inhibited hyphal transformation (Fig 5B), which is also consistent with the inhibition of hyphal morphological development by circDS-1 overexpression in static culture, indicating that high levels of circDS-1 have a negative effect on hyphal growth.

Finally, as virulence is inseparable with dimorphic transition, we performed spore-macrophage co-culture experiments. The spore virulence decreased under the down-regulation of circDS-1 (S5 Fig), and rescued to the wild type level under circDS-1 expressed Δ*TM020485* strain (S5 Fig), possibly due to defects in yeast transition. In summary, the high expression level of circDS-1 is essential for yeast cells, and its downregulation might disrupt yeast mor-phogenesis and the transition from mycelial to yeast forms, thus potentially impacting spore virulence.

Then we investigated the potential mechanisms by which circDS-1 participates in dimorphic transition. By additional RNA-seq of circDS-1^OE and circDS-1^RNAi mutants under mycelia and yeast conditions (S1 Table), we identified four mycelial modules and four yeast modules as phenotype-correlated modules via co-expression network (Fig 6A). Translation processes were significantly enriched under mycelium condition, particularly in ribosome assembly and

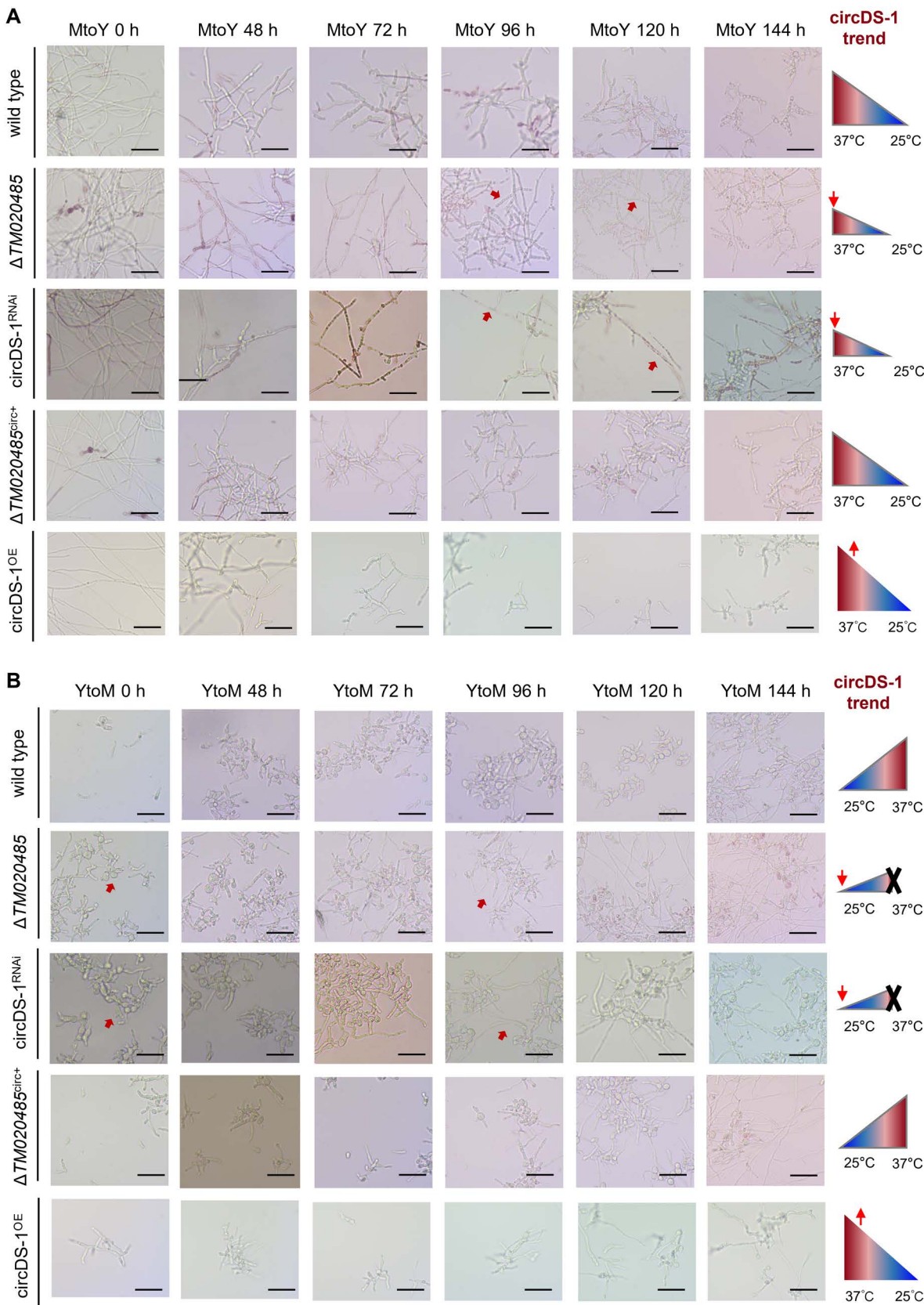

**Fig 5. circDS-1 regulates dynamic dimorphic transition. A,** Dynamic dimorphic transition from mycelia to yeast of wild type, Δ*TM020485*, circDS-1^RNAi, Δ*TM020485* rescued with circDS-1 (Δ*TM020485*^circ+) and circDS-1^OE strains. **B,** Dynamic dimorphic transition from yeast to mycelia of wild type, Δ*TM020485*, circDS-1^RNAi, Δ*TM020485* rescued with circDS-1 (Δ*TM020485*^circ+) and circDS-1^OE strains. The triangle diagram on the right represents the expression level of circDS-1 corresponding. Red arrows in figures indicate significantly different phenotypes between these strains during dimorphic transition. Scale bar of cell (black), 20 μm.

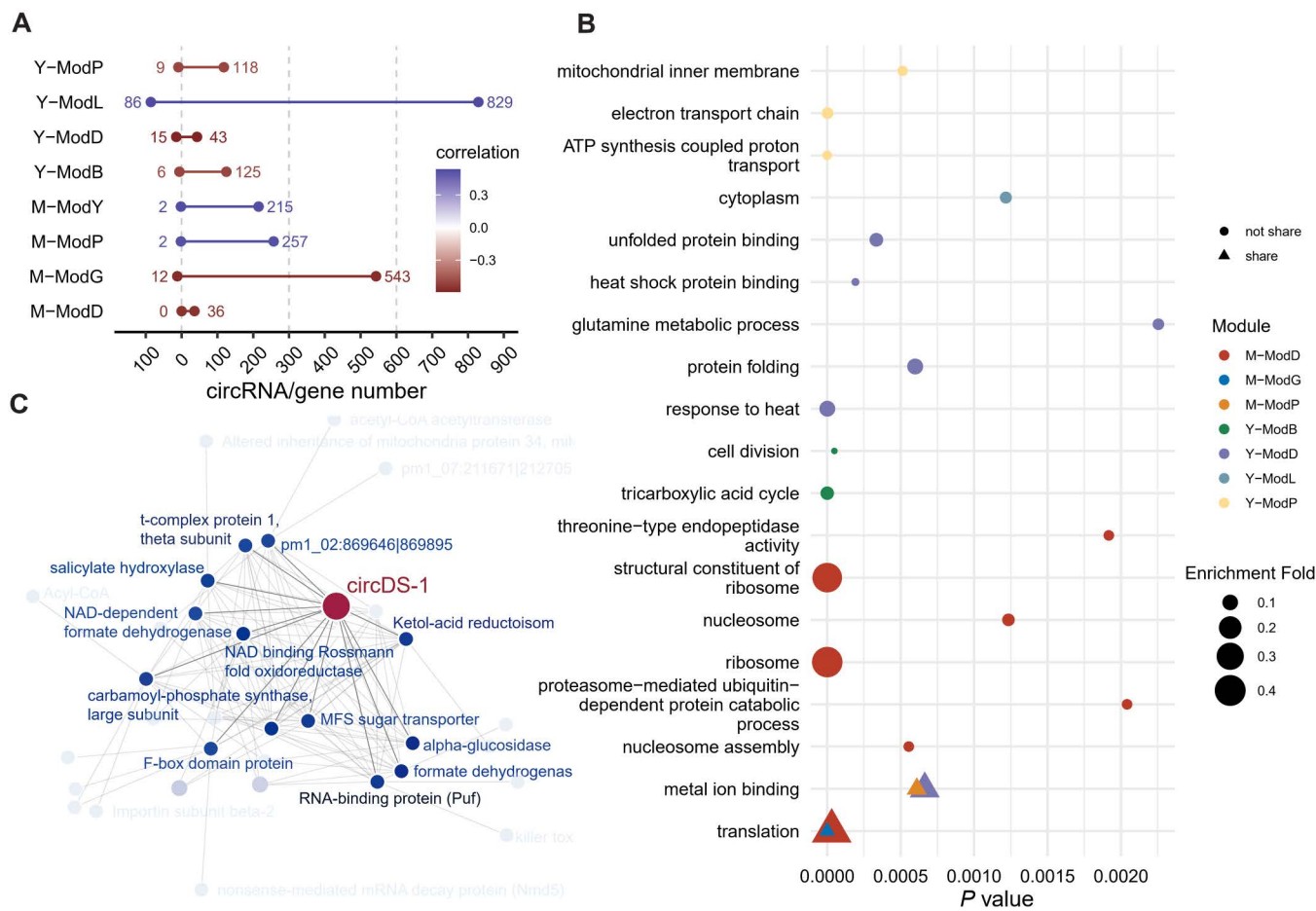

**Fig 6. Co-expression network of circDS-1 of mycelia and yeast. A**, Correlation coefficients and counts of circRNAs(left) and genes (right) in eight highly correlated phenotype-related modules in identified in co-expression network. **B**, Gene Ontology enrichment of genes in the eight phenotype-related correlated modules, respectively. The y-axis is the enriched GO terms. The size of the dot represents the enrichment fold. The shape of the points represents whether these GO terms are enriched simultaneously in different groups. **C**, Network visulazation of circDS-1 and highly co-expressed genes under yeast conditions. Genes directly co-regulated with circDS-1 are highlighted and labeled.

translation initiation (Fig 6B). In contrast, aerobic respiration, heat responses, and protein catabolic processes were prominent in yeast condition, suggesting an outstanding energy requirement for yeast morphology under 37°C (Fig 6B). Furthermore, genes co-expressed with circDS-1 in mycelia condition focused on nucleotide metabolism and transcription factor activity, indicating the abnormal expression levels of circDS-1 were related to global changes in transcriptome. CircDS-1 correlated genes in yeast condition were primarily focused on metabolism (Figs 6C and S6), indicating that metabolic dysregulation might contribute to the abnormal hyphae in yeast colonies when circDS-1 was silenced.

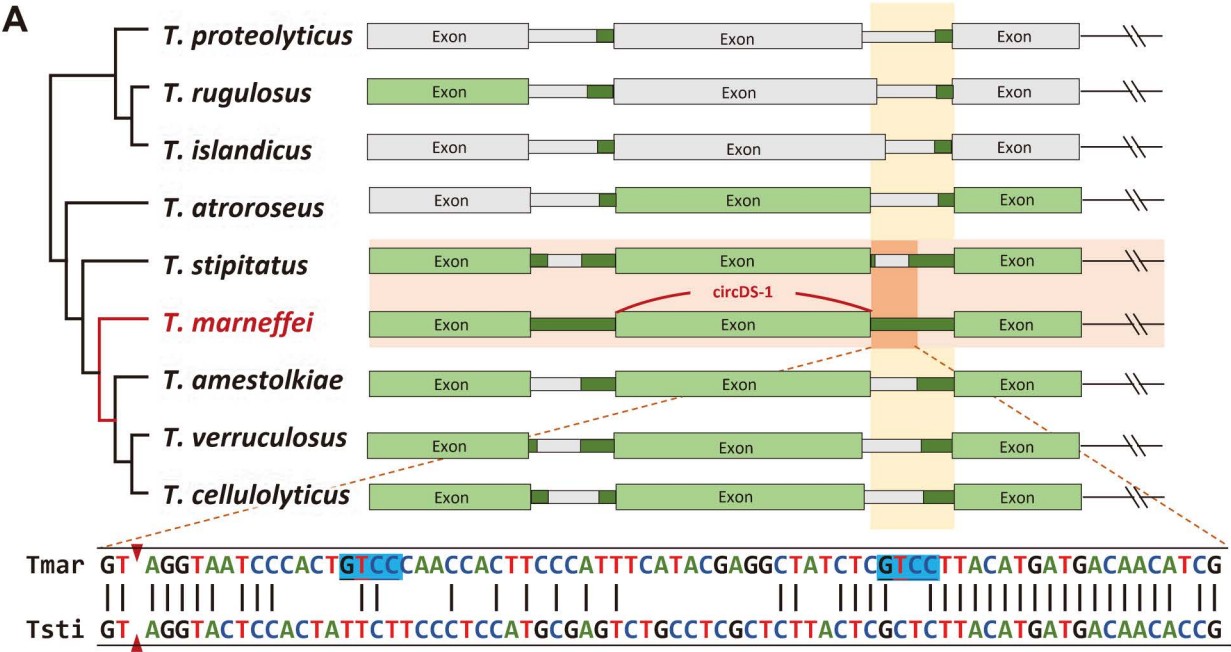

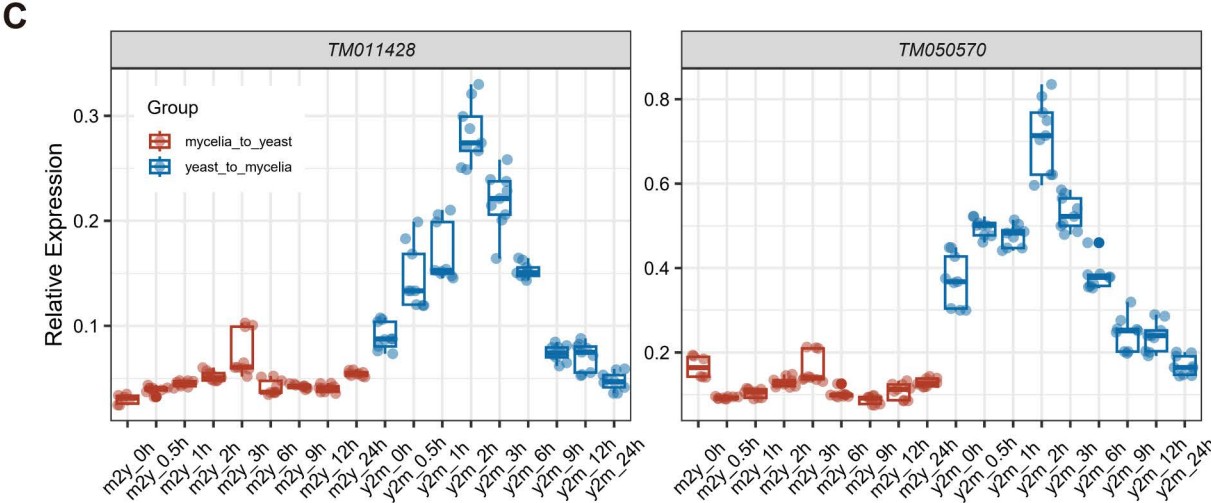

**Fig 7. Intronic hypermatation drives circularization of circDS-1. A**, Phylogenetic tree of *Talaromyces* and the corresponding gene structure of the circDS-1 parental gene. Green represents the sequence homology region of *T. marneffei* (alignment rate>60%). Gray represents sequence non-homologous regions of *T. marneffei* (alignment rate <60% or mismatch rate >20%). *T. marneffei* and its closest relative species *T. stipitatus* are marked by a horizontal orange box. The corresponding relationship between the 3' flanking sequences of circDS-1 is marked by a vertical yellow box. The detailed sequence alignment of the overlap between the two boxes are below. It is shown that the red triangle represents the exon-intron splicing site, the yellow area is the hyper-mutated region, and the blue is the binding site of KH-type RBP. **B**, Enriched motifs in circRNA upstream and downstream flanking sequences. **C**, Relative quantification of two KH-domain genes in dynamic transition between mycelia-to-yeast and yeast-to-mycelia. Three biological replicates and three technical replicates for each time point, and the relative expression was calculated using the $2^{-\Delta\Delta Ct}$ method, with actin as the internal reference gene.

## *T. marneffei*-specific flanking sequences contributes to circDS-1 generation

As circDS-1 was flanked by *T. marneffei*-specific sequences, we wondered whether the sequences regulated circDS-1 circularization. Initially, we assessed the conservation of the circDS-1 generation region. Despite high conservation in both genomic synteny and gene sequences across *Talaromyces* of circDS-1 itself, a species-specific intronic region was identified in the 3′-end flanking region of circDS-1 (Methods), which was close to but did not include the exon-intron splicing junctions or BSJ (yellow highlighted sequence in Fig 7A and Methods). We also examined the presence of this species-specific region in other well-known strains of *T. marneffei*, including 11CN-20-091 (GenBank: GCA_009556855.1), 11CN-03-130 (GenBank: GCA_009650675.1), ATCC 18224 (GenBank: GCA_009556855.1), and TM4 (GenBank: GCA_003971505.1). All four strains were found to harbor identical circDS-1 sequences along with their flanking regions, including the *T. marneffei*-specific region.

We then deleted the species-specific region while retaining the remaining flanking sequences, which resulted in the complete inhibition of circDS-1 circularization in *T. marneffei* (S7 Fig). Consequently, the expression of circDS-1 could not be detected in the closely related fungi *T. stipitatus* and *T. amestolkiae*, as they lack this region (S7 Fig). These results demonstrated the indispensable role of the species-specific intronic region in circDS-1 circularization, suggesting a potential evolutionary mechanism for circRNAs.

Given that complementary element-mediated circularization is widespread in humans [17], we thus examine the flank regions of circDS-1, and unexpectedly found no complementary sequences were detected in circDS-1 and even the majority of *T. marneffei* circRNAs (> 99%). We then preformed a systemic analysis of *trans*-elements in flanking region of all circRNAs, and identified significantly enriched RNA-binding protein (RBP) binding motifs in approximately 60% of circRNAs (Fig 7B), among which hnRNPs homologues and SR protein homologues emerging as the most prominent RBPs of *T. marneffei* at 21.9% and 37.8%, respectively (S2 Table). Over 10% of circRNAs with *T. marneffei*-specific flanking sequences exhibited the enrichment of well-known RBP motifs including homologues of FXR2, Vts1 and Hrp1. Notably, the *T. marneffei*-specific sequence flanking circDS-1 introduced two KH-type RBP binding motifs (Fig 7A), which may allow this region to bind to FXR2 homologues. During dimorphic transition of *T. marneffei,* KH-type RBPs exhibited transient and sudden expression changes in the initial hour during the transition from mycelia to yeast (Fig 7C), a pattern that mirrored the temporal expression trend of circDS-1 (S4 Fig). This suggests that KH-type RBP may potentially regulate its circularization by binding to the circDS-1 flanking region, akin to the role of human QKI as previously described [34].

## Discussion

Although causing millions of deaths, almost all the human pathogenic fungi are not obligate pathogens but from environments [4]. One of the essential challenges for these fungi to invade human is to survive from the high body temperature of host [8]. Thermally dimorphic fungi

evolve an intriguing adaptive mechanism by shifting morphologies between saprophytic mycelia at ambient temperature and pathogenic yeasts at mammalian body temperature [5]. Although thermally dimorphism is a rare trait across kingdom Fungi, thermally dimorphic fungi usually share extremely high similarity of protein-coding regions with the other fungi [35]. For example, the differences of protein-coding genes between *T. marneffei*, the only dimorphic fungus among over a thousand species in Eurotiales [35], and its closest relative *T. stipitatus* amount to less than 2% [22]. Therefore, non-coding elements are more likely attributed to thermally adaptation of *T. marneffei*, as well as the other thermally dimorphic fungi. In the current study, we identify a non-coding circular RNA molecular, named circDS-1, that contributes to thermally dimorphism of *T. marneffei*. As the first discovered functional circRNA in fungi, circDS-1, independent of its parental gene, facilitates the mycelia-to-yeast transition and sustains yeast morphology by participating in the regulation of heat stress responses and metabolic pathways. Although its parental gene is conserved within *Talaromyces*, the origination of circDS-1 is driven by the *T. marneffei*-specific intronic region, which introduces novel binding sites of splicing factor KH-type RBPs. Together, our findings demonstrate a novel molecular mechanism beyond protein-driven mechanisms on thermal adaptation, which may provide a flexible and cost-effective strategy for human pathogenic fungi in the context of dual adaptation to both environment and host.

Circular RNA is an ancient class of non-coding RNAs that are prevalent across Eukaryote and even detected in a few archaeal and bacterial species [20,36–41]. However, most of circRNA researches focus on animals and plants, especially human, which limits the understanding of circRNA evolution for the absent evidence from lower eukaryotic organisms. In our work, we demonstrate the atlas of high-quality circRNAs in *T. marneffei* by refining bioinformatic strategy for circRNA detection. The uneven species of circRNAs between yeast and mycelia (3,375 *vs.* 1,217) suggests that the unstable RNA process caused by higher temperature leads to more circRNAs, which aligns with higher eukaryotes that most circRNAs are by-products correlating with the fitness cost of splicing errors [42]. Nevertheless, in addition to the identified circDS-1, both more stably expressed circRNAs (expressed in all biological replicates under the same condition, 380 *vs.* 35, Fisher's Exact Test, $P = 9.08 \times 10^{-22}$) and significantly up-regulated circRNAs (357 *vs.* 3, Fisher's Exact Test, $P = 1.25 \times 10^{-45}$) under yeast than mycelium conditions exhibit marked yeast-enriched pattern, highlighting that some of these accidently-derived circRNAs may be under selection of thermal adaptation in *T. marneffei*. Thus, although calling for further studies in other lower eukaryotes, our current data support that the "chaos-to-function" model of long non-coding RNAs [43,44] and small RNA [45] could also be applied in the evolution of functional circRNA.

The damage response framework (DRF) states that organisms adopt specific cellular responses to cope with damage and stress, thereby achieving adaptation and survival [46]. In *T. marneffei*, the ability to undergo a dimorphic transition between yeast and hyphae forms is a response to environmental cues and host-induced damage. Previous studies have shown that protein-coding genes play important roles in DRF [47]. In this study, we found that circRNAs such as circDS-1 may regulate this process by regulating genes involved in translation, metabolism, and morphogenesis, and its deletion can significantly inhibit the virulence of *T. marneffei* in human macrophages. In addition, circDS-1 may participate in different regulatory pathways in hyphae and yeast phases, and this effect may be mediated by different expression levels or different thermodynamic molecular structures between 25°C and 37°C. The metabolic regulation at 37°C is likely related to cellular starvation stress, thereby promoting survival within macrophages. Although circDS-1 itself does not have coding ability, it may mediate the changes in the intracellular effective level of different protein molecules or complexes by binding to the latter, thus providing the possibility of post-translational regulation

and providing a more economical possibility for rapid regulation of effector levels in response to environmental cues.

The maintenance of exon-intron structure throughout the evolution of eukaryotes highly supports the evolutionary advantages of carrying introns [48]. Several benefits of intron have been illustrated, such as enhancing the complexities of transcripts and proteins, regulating RNA transportation, and harboring noncoding functional RNA genes [49,50]. However, rapid evolution in intronic regions is believed due to relaxed selective constrains rather than evolutionary advantages [51]. Although recent genome wide association studies (GWAS) and quantitative trait locus (QTL) analysis [52,53] have identified trait-associate single nucleotide polymorphisms (SNPs) in intron, whether introns are causally responsible for novel traits remains underexplored. Here, we unveil a novel mechanism of adaptive evolution exemplifying by circDS-1 in *T. marneffei* that mutation accumulated in intron drives the generation of functional circRNA. Our results support the advantage of intron as a reservoir of subtle variations driving *de novo* non-coding transcripts that fine-tune gene expression network during rapid adaption.

One of the limitations of this study is that although the function of circDS-1 may show a certain dose-dependent effect, we were unable to accurately overexpress circDS-1 to verify this possibility due to potentially complex circularization regulation in *T. marneffei*. Future studies on fungal circRNAs require more optimized overexpression and knockdown systems to explore the functions and mechanisms of circRNAs. Another limitation of this study is the lack of more reliable molecular markers to accurately track the dimorphic transition of *T. marneffei*. Although we used morphological features as the primary basis for determining yeast and hyphal morphology, this approach remains somewhat subjective and may not fully capture the dynamic nature of the transition. This highlights the need to develop robust molecular markers that can specifically identify and distinguish the various stages of the dimorphic transition. In future studies, it may be possible to address this limitation by integrating transcriptome-wide analyses, which could help track this critical process more accurately.

## Materials and methods

### Strains and culture conditions

The wild type *T. marneffei* strain PM1, which was isolated from a patient with culture-documented talaromycosis in Hong Kong [54], was used in this study. All strains (mutants and wild type) of *T. marneffei* were grown on Sabouraud Dextrose Agar (SDA, Oxoid, Cambridge, UK) at 25°C for seven days for conidia collection and phenotype observation, and at 37°C for seven days for observation. All strains of *T. marneffei* were cultured on Sabouraud Dextrose Broth (SDB, Oxoid, Cambridge, UK) at 25°C or 37°C for three days for total RNA extraction. The wild type *Talaromyces stipitatus* was purchased from CGMCC (China General Microbiological Culture Collection Center) and cultured on SDB under 25°C for 3 d for RNA extraction. The wild type *Talaromyces amestolkiae* was purchased from CICC (China Center of Industrial Culture Collection) and cultured on SDB under 28°C for 3 d for RNA extraction.

### Cell culture

The human monocytic cell line THP-1 (purchased from Pulizhicheng, Beijing, China) was cultured at 37°C in Roswell Park Memorial Institute 1640 medium (Gibco, Carlsbad, CA) supplemented with 10% fetal bovine serum (Gibco, Carlsbad, CA). THP-1 cells were stimulated with 100 ng/mL phorbol 12-myristate 13-acetate (Beyotime Biotechnology, Shanghai, China) for 24 h for differentiating into macrophages [55].

## RNA extraction

Total RNA of strains was extracted using the E.Z.N.A. fungal RNA kit (Omega Bio-Tek), following the manufacturer's instructions with DNase I digestion to eliminate genomic DNA. The RNA concentration was measured using Nanodrop (Thermo Fisher Scientific Inc. USA), and the quality was assessed using $Qsep_1$ (BiOptic, Inc., New Taipei City, Taiwan). RNA with a RIN > 8 was selected and stored at -80°C until downstream preparation.

## Illumina library construction

Ribosome RNA was depleted from total RNA as previously described [56]. The rRNA-depleted total RNA was prepared using the VAHTS Universal V6 RNA-seq Library Prep Kit for Illumina (Nanjing Vazyme Biotech Co., Ltd. China) according to the manufacturer's protocol. The processed RNA was fragmented into short fragments of 150–250 nt using fragmentation buffer, followed by reverse transcription into cDNA with random hexamer primers. The second strands were synthesized with dUTP buffer mix to generate strand-specific libraries. Different adapters were used for pooled sequencing. Final PCR products were purified with 0.9×DNA Clean Beads (Nanjing Vazyme Biotech Co., Ltd. China) and sequenced on Illumina Novaseq (Novogene, Beijing, China) with paired-end 150 bp reads. Each library generated approximately 150 to 200 million pairs of reads (S1 Table).

## Nanopore library construction

Full-length libraries were prepared as previously described from total RNA [24]. In brief, rRNA depletion and RNase R treatment were used for circRNA enrichment and then all treated RNA was reverse transcribed into cDNA with special anchorX-$N_6$ primers, followed by the addition of a poly(A) tail and synthesis of the second strand with anchorY-$T_{24}$ primers. The dsDNA was amplified using paired primers anchorX and anchorY, and the purified PCR products were sequenced on the Oxford Nanopore Platform (GrandOmics, Beijing, China). Each library generated over 10 million reads (S1 Table).

## Qualitative and quantitative validation of circRNA

Qualitative validation was performed by designing forward and divergent primers (S3 Table) according to actual or predicted sequences of ten randomly selected circRNAs as previously described [57]. Briefly, total RNA of mycelium and yeast conditions was reverse transcribed into cDNA by HiScript III 1st Strand cDNA Synthesis Kit (Nanjing Vazyme Biotech Co., Ltd. China) with genome DNA wiped according to the manufacturer's instructions, respectively. The linear region and back-splicing junction (BSJ) region were amplified using 2×Phanta Flash Master Mix (Nanjing Vazyme Biotech Co., Ltd. China), and the products of forward primers were observed in genomic DNA (gDNA) and cDNA of mycelia and yeast, while the products of divergent primers were observed only in cDNA due to the specific circular structures. To validate the back-splicing events, the products of divergent primers were purified and sequenced by Sanger sequencing. For quantitative validation, the same reverse transcription as above was performed, and real-time quantitative PCR was performed using qPCR primers (S3 Table) by Taq Pro Universal SYBR qPCR Master Mix (Nanjing Vazyme Biotech Co.,Ltd. China) to quantify circRNA and cognate linear RNA in mycelium and yeast conditions with *actin* as the reference gene. Each qPCR experiment was performed with three technological replicates and RQ (Relative quantification) was calculated by $2^{-\Delta\Delta Ct}$ methods.

## CircRNA candidate identification

CircRNAs were identified from clean RNA-seq of Illumina platform and Nanopore platform by CIRI2 [58] (NGS-only) and CIRI-long [26] (TGS-only) using PM1 genome and annotation (PRJNA967827), respectively. In NGS&TGS, only overlapping circRNAs in the results of CIRI2 and CIRI-long were retained. In NreT-seq, all back-splice junctions identified in Nanopore full-length reads were re-checked using Illumina short reads via CIRIquant using default parameters [59]. BSJs that were supported by Illumina sequencing data in result of CIRIquant were retained as the final results of NreT-seq, which rescued those BSJs that were not identified by CIRI2 but were supported by Illumina sequencing or were mapped inaccurately in Nanopore sequencing.

## Composition annotation of full-length circRNA

The full-length sequences of circRNAs were assembled using the aligned circRNA fragments provided by CIRI-long using default parameters, and the most frequent fragments were considered as the optimal exons of circRNAs (circ-exon). All circ-exons were then intersected with the genomic GTF file to precisely locate the internal composition of full-length circRNAs using in-house scripts.

## Flanking motifs identification of circRNA

Flanking regions were defined as 500 nt upstream or downstream of circRNA and aligned to genome using BLASTn [60] for complementary element scanning. For motif scanning, only ±20 bp around back-splicing sites of circRNAs or exon of linear transcripts were used to avoid interference from further exons sequences as the average length of introns was 76 bp in *T. marneffei*. Software STREME [61] was used for motif identification and software Tomtom [62] was used for motif comparison against RNAcompete data [63] and CISBP-RNA Database [64]. The homology genes of identified RBPs were examined using tblastn of software BLAST against *T. marneffei* genome with a parameter e-value of $10^{-3}$, and genes of *T. marneffei* with pident > 30 were considered as homologous genes.

## Expression analysis of circRNAs

Differential expressed circRNAs were identified by CIRIquant with default parameters. Differential expressed genes were identified by R packages DESeq2 [65] and gene with |log$_2$(fold change)| > 1 and false discovery rate (FDR) < 0.05 were identified as DE genes. In differential transcription analysis, DE ratio circRNAs were identified by Riborex scripts [66] with modulation that using transcript type (linear or circular) as the covariate. Significant Score (DScore) of circRNAs were calculated as

$$DScore = \frac{FDR_{circ}}{CPM_{circ} \times Ratio_{circ} \times \log_2\left(FC_{circ}\right) \times \log_2\left(FC_{ratio}\right) \times \log_2\left(FC_{gene}\right)}$$

## Construction of circDS-1 overexpression vector

Considering that splice sites of introns might affect circularization, we extended the insertion sequences to upstream intron 3 and downstream part of exon 6 in ectopic overexpression vector. The circDS-1 overexpression plasmid pOEcircDS-1 were constructed based on the backbone of plasmid pSilent-1 using promoter and terminator of *Aspergillus nidulans trpC* gene to overexpress circDS-1 fragment. The region of trpC(p),

intron3-exon4-intron4-circDS-1-intron5-parital exon6, trpC(t) and pSlient-1 backbone were cloned by PCR and assembled by ligase-independent multi-fragment cloning technique using ClonExpress MultiS One Step Cloning Kit (Nanjing Vazyme Biotech Co.,Ltd. China). The propagation of pOEcircDS-1 was performed by chemical transformation of *Escherichia coli* DH5α cells (Nanjing Vazyme Biotech Co.,Ltd. China) according to the manufacturer's instruction.

## Construction of circDS-1 knock down vector

The 53 bp shRNA#1-3 sequence used for circDS-1 silencing was designed across the BSJ site of circDS-1 that is distinct from the cognate linear transcript. Taking the BSJ junction as the 0 coordinate, the sequence coordinates of shRNA#1 was -11/+10, the sequence coordinates of shRNA#2 was -15/+6, and the sequence coordinates of shRNA#3 was -6/+15. The plasmids of shRNA#1-#3 were seamlessly cloned by ClonExpress II One Step Cloning Kit (Nanjing Vazyme Biotech Co.,Ltd. China) according to manufacturer's instruction. The propagation of circDS-1$^{RNAi}$#1-#3 was performed by chemical transformation of *Escherichia coli* DH5α cells (Nanjing Vazyme Biotech Co.,Ltd. China) according to the manufacturer's instruction.

## Construction of *TM020485* deletion vector

The *TM020485* deletion vector was constructed based on plasmid pSD-1 using upstream 1,010 bp and downstream 523 bp of *TM020485* as homology arms. The homology arms were cloned into the upstream and downstream G418 resistance genes of using ClonExpress II One Step Cloning Kit (Nanjing Vazyme Biotech Co.,Ltd. China) according to manufacturer's instruction. The propagation of Δ*TM020485* was performed by chemical transformation of *Escherichia coli* DH5α cells (Nanjing Vazyme Biotech Co.,Ltd. China) according to the manufacturer's instruction.

## Construction of insertion sequences deletion vector

The *T. marneffei*-specific sequences deletion vector was constructed based on plasmid pSilent-1, using the same expression cassette of circDS-1 overexpression vector except for removing intronic *T. marneffei*-specific region sequences "AATCCCACTGTCCCA ACCACTTCCCATTT

CATACGAGGCTATCTCGTCC" with remain flanking sequences retained. Corresponding fragments were amplified from the genome and backbone plasmids using corresponding primers, and each fragment was assembled using seamless cloning technology. The assembled plasmid was verified by Sanger sequencing. Since both vectors were expressed in the same background strain Δ*TM020485*, the two (Δ*TM020485*$^{circDS-1-IK}$ and Δ*TM020485*$^{circDS-1+}$) have only differences in the circularization flanking sequences in the regulation of circDS-1 expression.

## Protoplast preparation and transformation

The WT or Δ*TM020485* mutant strains were transformed with plasmids based on the protocol previously described with several modifications [67]. Briefly, the strains were cultured on SDA medium at 25°C for 7 days. Spores were collected by fine toothpick scraping, washed twice and incubate for germination in an incubator at 37 °C, 150 rpm/min for 40 h. The germinated spores were centrifuged at 4200×g for 5 min, and the supernatant was removed as much as possible. Germinated spores were added into 40 mL of enzymatic hydrolysis solution (β-Glucuronidase and Driselase, Sigma) at 37°C, 80rpm until 50% protoplasts are produced by spores. The Miracloth (Calbiochem, Germany) was used to filter cell debris and the

protoplasts were washed twice. Clean protoplasts were stored in -80°C until transformation. For transformation, a total of 5 μg plasmid were mixed with $1\times10^6$ protoplasts and incubate on ice for 30 min. Then 1 mL PTC (60% PEG6000, 100 mM $Ca^{2+}$) was added into the mixture and incubated under room temperature for 60 min. Transformants were cultured on corresponding resistance medium at 37°C for mutant selection and validated by whole genome sequencing.

### Dynamic dimorphic transition

After culturing in SDB at 25°C or 37°C for 3 d, the strain was switch to another temperature (such as 25°C transposed to 37°C). Smear observation was performed at 0, 12, 24, 36, 48, 72, 96, 120, 144 and 168 h after the temperature transition. RT-qPCR were performed at 0, 0.5, 1, 2, 3, 6, 9,12 and 24 h after the temperature transition.

### Macrophage co-culture

THP-1 macrophages were infected with conidia of different strains as indicated at MOI 1:10. After 2 h of incubation at 37 °C, extracellular conidia were removed and infected monolayers were washed by PBS for three times. One group was directly subjected to colony-forming unit (CFU) counting, and the other group was cultured for an additional 24 h before CFU counting to further confirm the spore resistance within macrophages as previously described [68]. Each group was conducted with three biological replicates.

### Construction of co-expression network and function enrichment

R packages WGCNA [69] was performed in each condition to construct OE-WT-KD networks and the soft threshold recommended by the package was used. Modules with *P* value < 0.05 and correlation coefficient > 0.5 were considered to be significantly related to phenotypes and were retained for functional annotation. The R package ClusterProfiler [70] was used for all functional enrichment analyses, and in-house annotation files were used as previously described [56]. The top 30 correlated genes of circDS-1 were selected for network visualization using the R package networkD3 (http://christophergandrud.github.io/networkD3/).

### Genome-wide synteny and collinearity analysis of *Talaromyces*

The genome sequences and annotation of eight species (*T. amestolkiae* (ASM189636v1), *T. atroroseus* (ASM190759v1), *T. cellulolyticus* (ASM157146v2), *T. islandicus* (PIS), *T. stipitatus* (JCVI-TSTA1-3.0), *T. verruculosus* (ASM130527v1), *T. proteolyticus* (Talpro1) and *T. rugulosus* (ASM1336875v1)) were downloaded in National Center for Biotechnology Information (NCBI). The longest transcript of each species was extracted from the genome annotation file, and the protein sequence corresponding to the longest transcript was extracted using GFFread software [71]. Genome-wide synteny analysis was performed using JCVI software as previously described [72].

### Conservation analysis of circRNA and flanking region

For CDGs, we employed the OrthoFinder [73] software and extracted lineage-specific orphan genes by isolating those unique to *T. marneffei* as identified by the OrthoFinder. For conservation score, RepeatModeler [74] software was employed for *de novo* identification and RepeatMasker [75] software was then used to soft-mask the repetitive sequences in the genome. Subsequently, LASTZ software facilitated pairwise whole-genome comparisons and the alignment results were input into the MULTIZ software. Finally, the PhastCons module of PHAST [76] software

was utilized to calculate conservation scores. For flanking sequences, the genomic sequences of *T. marneffei* and *T. stipitatus* were subjected to whole-genome comparison using the LASTZ software. CircRNA flanking sequences, consisting of 50 bp upstream and downstream, were extracted and classified into exon or intron types and the conservation scores were calculated by PhastCons module. *T. marneffei*-specific flanking sequences were identified by alignment with the *T. stipitatus* genome using the BLAST software. Sequences with matched length less than 60% or mismatch exceeding 20% were considered as non-conservative flanking sequences.

## Time-course analysis of dimorphic transcriptome

The expression matrix of KH-type RBPs during dimorphic transition was obtained from the National Center for Biotechnology Information BioProject database under the accession PRJNA970557.

## Data analysis and visualization

All data analyses were performed using R v4.1.2. Visualizations were performed using ggplot2 v3.3.6 except mentioned separately.

## Statistical tests

All statistical tests are completed in R v4.1.2, and the corresponding test methods and statistics are marked in the corresponding text.

## Supporting information

**S1 Fig. Agarose gel electrophoresis of randomly selected circRNAs.** Divergent primers could amplify BSJ of circRNAs, thus products could be detected in mycelia and yeast cDNA (McDNA and YcDNA) but not genomic DNA (gDNA). Forward primers were designed as controls to exclude trans rearrangement events at the genomic level that may mimic BSJ sequences. BSJ is represented by a red triangle, and the red circle represents the amplification product of "BSJ+full length*n", which is different from linear RNA.
(TIF)

**S2 Fig. Gene Ontology terms enriched in circRNAs with non-conserved T. marneffei-specific flanking sequences.** The y-axis is the enriched GO terms. The size of the dot represents the identified gene counts.
(TIF)

**S3 Fig. Validation of the top 5 circRNAs by forward/divergent PCR and Sanger sequencing.** The products of BSJs are indicated by red triangles, while the red circles represent the amplified products that occur specifically in circRNAs but not in linear RNAs.
(TIF)

**S4 Fig. Relative quantification of circDS-1 in dynamic transition between mycelia-to-yeast and yeast-to-mycelia.** Three biological replicates and three technical replicates for each time point, and the relative expression was calculated using the $2^{-\Delta\Delta Ct}$ method, with actin as the internal reference gene.
(TIF)

**S5 Fig. After additional incubation for 2 and 24 hours following co-culture challenge of macrophages, colony-forming units were counted for Δ*TM020485*, wild type, and Δ*TM020485* rescued with circDS-1 (Δ*TM020485*^circ+^) spores.**
(TIF)

**S6 Fig.  Gene Ontology enrichment of genes that highly correlated with circDS-1 in mycelia and yeast.** The y-axis is the enriched GO terms. The size of the dot represents the identified enrichment fold.
(TIF)

**S7 Fig.  Relative quantification of circDS-1 in *T. marneffei* strains with specific region knockouts and in closely related species lacking this region. A,** Ct values of reference gene actin and circDS-1 in specific region knockouts (IK), Δ*TM020485*, Δ*TM020485* rescued with circDS-1 (Δ*TM020485*circ+) and wild type strains. **B,** Amplification signals of circDS-1 in *T. stipitatus* and *T. amestolkiae.*
(TIF)

**S1 Table.  Sequencing information for NGS and TGS RNA-seq of mycelia and yeast samples of WT strains and mutant strains.**
(CSV)

**S2 Table.  Enriched motifs in circRNA flanking sequences and predicted RNA-binding proteins (RBPs) in *T. marneffei*.**
(TXT)

**S3 Table.  Primers utilized for circRNA validation and quantification in this study.**
(XLSX)

## Author contributions

**Conceptualization:** Xueyan Hu, Minghao Du, Ence Yang.

**Data curation:** Xueyan Hu, Minghao Du.

**Formal analysis:** Xueyan Hu, Minghao Du.

**Funding acquisition:** Ence Yang.

**Investigation:** Xueyan Hu, Minghao Du, Changyu Tao, Juan Wang, Yun Zhang, Yueqi Jin.

**Methodology:** Xueyan Hu, Minghao Du, Changyu Tao.

**Project administration:** Minghao Du, Changyu Tao, Ence Yang.

**Resources:** Minghao Du, Changyu Tao.

**Software:** Xueyan Hu, Minghao Du, Changyu Tao.

**Supervision:** Ence Yang.

**Validation:** Juan Wang, Yun Zhang, Yueqi Jin.

**Visualization:** Xueyan Hu.

**Writing – original draft:** Xueyan Hu.

**Writing – review & editing:** Xueyan Hu, Minghao Du, Changyu Tao, Juan Wang, Yun Zhang, Yueqi Jin, Ence Yang.

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
