## [Decision Letter · Decision Letter 0]

15 Jan 2025

PGENETICS-D-24-01296

Species-specific circular RNA circDS-1 enhances adaptive evolution in Talaromyces marneffei through regulation of dimorphic transition and virulence

PLOS Genetics

Dear Dr. Yang,

Thank you for submitting your manuscript to PLOS Genetics. After careful consideration, we feel that it has merit but does not fully meet PLOS Genetics's publication criteria as it currently stands. Therefore, we invite you to submit a revised version of the manuscript that addresses the points raised during the review process.

Please submit your revised manuscript within 30 days Feb 14 2025 11:59PM. If you will need more time than this to complete your revisions, please reply to this message or contact the journal office at plosgenetics@plos.org. Please include the following items when submitting your revised manuscript:

We look forward to receiving your revised manuscript.

Kind regards,

Benjamin Schwessinger

Academic Editor

PLOS Genetics

Geraldine Butler

Section Editor

PLOS Genetics

Aimée Dudley

Editor-in-Chief

PLOS Genetics

Anne Goriely

Editor-in-Chief

PLOS Genetics

**Journal Requirements:**

At this stage, the following Authors/Authors require contributions: Xueyan Hu, Minghao Du, Changyu Tao, Juan Wang, Yun Zhang, Yueqi Jin, and Ence Yang. Please ensure that the full contributions of each author are acknowledged in the "Add/Edit/Remove Authors" section of our submission form.

The list of CRediT author contributions may be found here: https://journals.plos.org/plosgenetics/s/authorship#loc-author-contributions

- ® on page: 38 line 644.

5) We notice that your supplementary Figures are included in the manuscript file. Please remove them and upload them with the file type 'Supporting Information'. Please ensure that each Supporting Information file has a legend listed in the manuscript after the references list.

Potential Copyright Issues:

i) Please confirm (a) that you are the photographer of 4, or (b) provide written permission from the photographer to publish the photo(s) under our CC BY 4.0 license.

7) Please amend your detailed Financial Disclosure statement. This is published with the article. It must therefore be completed in full sentences and contain the exact wording you wish to be published.

2) State what role the funders took in the study. If the funders had no role in your study, please state: "The funders had no role in study design, data collection and analysis, decision to publish, or preparation of the manuscript.".

**Reviewers' comments:**

Reviewer's Responses to Questions

Reviewer #1: Comments for the Authors

Using a new sequencing strategy (NreT-seq), the authors investigate the conservation of circRNAs in T. marneffei. Through differential expression analysis, they find that circRNAs are involved in the regulation of the dimorphic transition of T. marneffei, with circDS-1 exhibiting the highest fold change in differential transcription. They further discover that circDS-1 is less expressed in mycelia and more expressed in yeast, and that down-regulation of circDS-1 inhibits the mycelia-to-yeast transition. Moreover, the T. marneffei-specific flanking sequence contributes to the generation of circDS-1, which may utilize an “intron definition” splicing model. Overall, the data are of high quality and judiciously interpreted. I have only a few minor comments that might help further clarify this important finding.

Specific comments:

1. As shown in Figure 4B, the expression level of circDS-1 in OE#rep1 was higher than in OE#rep2 and rep3 (labeled rep2), and the morphogenesis of OE#rep1 was noticeably different from that of OE#rep2 and rep3. The morphology of OE#rep1 resembled mycelia more closely. Does further overexpression of circDS-1 lead to changes in mycelial morphology?

2. The relative quantification (RQ) of circDS-1 in mycelia and yeast is presented. In my view, the authors should test the differential expression of circDS-1 during the transition from mycelia to yeast or from yeast to mycelia.

3. In Figure 4, the authors should clarify the meaning of the purple arrow.

4. The authors should illustrate the dynamic dimorphic transition from mycelia to yeast and the mycelia of the wild type of circDS-1OE.

5. The resolution of the image in Figure 5 is low.

6. The authors stated “We then deleted the species-specific region while retaining the remaining flanking sequences, which resulted in the complete inhibition of circDS-1 circularization in T. marneffei.” The sequence should be illustrated, and the results should be provided, as they are important evidence to support the conclusion.

Reviewer #2: The data of circDS-1 in TM-infected mouse model is expected.

Reviewer #3: This study is robust and valuable, providing new evidence regarding circRNA. However, the evaluation of genotype and phenotype characteristics appears to be insufficient. Would it be feasible to assess pathogenicity using simple animal models, such as Galleria mellonella larvae? This approach could further enhance the significance of the research.

Another point of consideration is the morphological transformation of Talaromyces marneffei, particularly in the yeast phase, which seems to remain unclear and difficult to distinguish from its morphological characteristics. The development of a tool to confirm yeast phase specific markers is crucial. If such confirmation cannot be performed in the laboratory, it would be important to discuss this limitation.

Furthermore, the current understanding of the pathogenicity of Talaromyces marneffei has shifted. Additional discussion on the Damage Response Framework (DRF) is necessary, as the findings in this study provide significant support for this concept.

**Have all data underlying the figures and results presented in the manuscript been provided?**

Reviewer #1: None

Reviewer #2: Yes

Reviewer #3: Yes

PLOS authors have the option to publish the peer review history of their article (what does this mean? ). If published, this will include your full peer review and any attached files.

**Do you want your identity to be public for this peer review?** For information about this choice, including consent withdrawal, please see our Privacy Policy .

Reviewer #1: No

Reviewer #2: No

Reviewer #3: No

**Figure resubmission:**
---

## [Editor Report · Decision Letter 1]

12 Feb 2025

Dear Dr Yang,

We are pleased to inform you that your manuscript entitled "Species-specific circular RNA circDS-1 enhances adaptive evolution in *Talaromyces marneffei* through regulation of dimorphic transition" has been editorially accepted for publication in PLOS Genetics. Congratulations!

Yours sincerely,

Benjamin Schwessinger

Academic Editor

PLOS Genetics

Geraldine Butler

Section Editor

PLOS Genetics

Aimée Dudley

Editor-in-Chief

PLOS Genetics

Anne Goriely

Editor-in-Chief

PLOS Genetics

Comments from the reviewers (if applicable):

**Data Deposition**

http://datadryad.org/submit?journalID=pgenetics&manu=PGENETICS-D-24-01296R1

**Press Queries**

---

## [Editor Report · Acceptance letter]

PGENETICS-D-24-01296R1

Species-specific circular RNA circDS-1 enhances adaptive evolution in *Talaromyces marneffei* through regulation of dimorphic transition

Dear Dr Yang,

We are pleased to inform you that your manuscript entitled "Species-specific circular RNA circDS-1 enhances adaptive evolution in *Talaromyces marneffei* through regulation of dimorphic transition" has been formally accepted for publication in PLOS Genetics! Your manuscript is now with our production department and you will be notified of the publication date in due course.

With kind regards,

Zsofia Freund

PLOS Genetics

On behalf of:
